# Hormonal and Reproductive Factors in Relation to Cardiovascular Events in Women with Early Rheumatoid Arthritis

**DOI:** 10.3390/jcm12010208

**Published:** 2022-12-27

**Authors:** Antonia Boman, Heidi Kokkonen, Ewa Berglin, Gerd-Marie Alenius, Solbritt Rantapää-Dahlqvist

**Affiliations:** Department of Public Health and Clinical Medicine/Rheumatology, Umeå University, 90187 Umeå, Sweden

**Keywords:** early rheumatoid arthritis, cardiovascular events, cardiovascular disease, risk factors, hormonal factors, reproductive factors

## Abstract

Hormonal and reproductive factors affect the risk for cardiovascular events (CVE) in the general population. Although the risk of CVE is increased in rheumatoid arthritis (RA), the knowledge about the impact of hormonal factors for CVE in RA is sparse. Female postmenopausal patients ≤80 years with early RA were consecutively included in this observational study (*n* = 803) between 1 January 1996 until 31 December 2017. Questionnaires regarding hormonal factors were distributed from the index date. Data regarding CVE were obtained from the Swedish National Health Register and Cause of Death Register. Associations between CVE and hormonal factors were analyzed using Cox proportional hazard regression. Of the postmenopausal women, 64 women had a CVE after RA onset. The time period from menopause to RA onset was significantly longer for CVE cases with higher proportion of postmenopausal women. In Cox proportional hazard regression models, years from last childbirth and multiparity were associated with higher CVE risk. Adjustments for traditional risk factors did not affect the results except for hypertension. RA onset after menopause and a longer duration from menopause until onset increased the CVE risk. Multiparity was associated with higher CVE risk whilst oral contraceptives decreased the risk. These results can contribute to identification of high-risk patients for CVE beyond traditional risk factors.

## 1. Introduction

Patients with rheumatoid arthritis (RA) have an increased risk of cardiovascular disease (CVD), which contributes to the largest proportion of excess mortality compared with the general population, particularly ischemic heart disease, not fully accounted for by traditional risk factors [1,2,3]. Other CVD manifestations are atherosclerosis, arterial stiffness, coronary arteritis, congestive heart failure, and valvular disease [4,5]. Suggested underlying mechanisms linking the increased risk of CVD to RA besides inflammatory-related mediators, include alterations of lipoproteins, increased oxidative stress, and endothelial dysfunction [6,7]. A number of risk factors are in common between development of RA and of CVD per se such as smoking, air pollution, overweight, increased BMI, and increased CRP, ACPA [8].

Furthermore, there is a difference in traditional risk factors, in general, between men and women, where results increasingly suggest that several reproductive factors associate with later-life CVD [9,10].

In relation to the development of RA, hormonal factors have been analyzed in a number of studies [11,12,13]. The increased prevalence of RA in women compared to men, with a peak incidence that coincides with menopause [11], has led to the hypothesis that hormonal and reproductive factors play an important role in the development of RA [11,14]. However, clinical and experimental studies have shown conflicting results regarding the pathogenic and etiological role of female hormones in RA [11,14]. There are also consistent results suggesting that estrogens have a protective role in disease activity and progression of RA [15,16]. For example, most patients experience disease remission during pregnancy when levels of female sex hormones are high but a relapse during the postpartum period [15,16]. Interestingly, estrogens have both pro- and anti-inflammatory effects that may explain their paradox in RA pathogenesis. While estrogens are potent inducers of B cell survival and antibody production, they also decrease the production of key cytokines for RA pathogenesis, such as IL-6, TNFα, and IL-1β [17]. The peak incidence of onset in females coincides with menopause, when the ovarian production of sex hormones drops markedly [18].

Other hormones, such as androgens, progesterone or prolactin, also exert effects on the immune system. The postmenopausal stage, an early age at menopause, the post-partum period, and the use of anti-estrogen agents are associated with RA onset. All these phenomena have in common an acute decline in ovarian function and/or in estrogen bioavailability [11].

Several hormonal and reproductive factors in women of the general population have been shown to associate with the risk of developing later-life CVD, although findings are diverse and inconsistent [10,19,20,21,22,23]. An increased risk for CVD later in life has been associated with early menarche and menopause [10,19,20], parity (i.e., number of childbirths) [20,21], childbirths at an early age [22], and the use of oral contraceptives [23]. Protective hormonal and reproductive factors for CVD have also been reported, for example, in women who had breastfed earlier in life, suggesting long-term health benefits for the women [24]. The conflicting results can partly be explained by differences in treatment and differences in populations over time regarding hormonal and reproductive factors. To evaluate total estrogen exposure, several indices have been constructed [25] and a majority of studies have defined this as the reproductive lifespan, i.e., the time period between menarche and menopause [25,26,27]. A longer reproductive lifespan has also been suggested to be protective against CVD [25,28]. Conclusive findings regarding the associations of hormonal factors and CVD in patients with RA are however lacking.

In the light of studies concerning hormonal and/or reproductive factors for both CVD per se and for RA, we have analyzed the associations between hormonal and reproductive factors, and RA disease-related factors for the risk of developing CVE in women with early RA. We have, as one of the first studies focused on factors previously analyzed for women of the general population, to explore the impact of these factors in women with RA for affection with CVE.

## 2. Materials and Methods

### 2.1. Subjects

Patients from the northern region of Sweden (rheumatology departments at five hospitals) with early RA (symptoms ≤12 months) fulfilling the 1987 American Rheumatism Association (ARA) classification criteria for RA [29] were consecutively included into the study and registered in the Swedish Rheumatology Quality Register [30] since 1 January 1996. This study included women from the early RA cohort from 1 January 1996 until 31 December 2017. The patients were followed from the index date until death or the end of the study.

Data regarding Disease Activity Score (DAS28) [31] including swollen (SJC) and tender joint count (TJC), patient’s global assessment (visual analogue scale) and erythrocyte sedimentation rate (ESR; mm/hour) were assessed at inclusion (index date) and at 6, 12, 18, and 24-months during follow-up continuously. The patients were treated with the aim of achieving remission/low disease activity [32] identified by the patient’s physician. The treatment constituted of conventional disease-modifying antirheumatic drugs (csDMARDs; azathioprine, chloroquine, cyclosporine, leflunomide, methotrexate, myocrisine, and sulfasalazine), corticosteroids, and biological (b)DMARDs (abatacept, adalimumab, anakinra, certolizumabpegol, etanercept, golimumab, infliximab, rituximab, and tocilizumab). For this study, data were presented from index date for the first 24 months for csDMARDs and bDMARDs.

Self-administered questionnaires concerning hormonal and reproductive factors, such as age at menopause and menarche, number of childbirths, abortion or miscarriage, duration of breastfeeding, hormonal birth control, and hormonal replacement therapy (HRT) were continuously sent out to female patients ≤80 years between two to five years after inclusion into the early arthritis cohort. In total, 1239 patients with early RA were eligible for the study but 40 were excluded as they had deceased after index date before receiving the questionnaire, six had developed dementia, another eight were not possible to identify due to wrong addresses, incorrect personal identification numbers or emigration and five did not want to participate. Of the remaining 1185 patients diagnosed with early RA, 928 (78.6%) patients completed the questionnaires and were included in the study (Figure 1). Further, 125 premenopausal patients who had answered the questionnaires were excluded from the analyses, since we have focused our analyses on postmenopausal women. We included women with natural or surgical (oophorectomy; *n* = 3) menopause. Reproductive lifespan was calculated as the time period in years between menarche and menopause. Information regarding smoking habits and body mass index (BMI kg^2^/m) was also asked for. The patients were classified either as being a non-smoker or ever-smoker (past or current).

Data regarding cardiovascular events (CVE) was obtained from the Swedish National Health Register and Cause of Death Register and were defined as hospital admissions or death as the underlying cause, e.g., ischemic stroke (ICD-10 codes I63.0–6, I63.8–9, I64.9, 165.0–3, 165.8–9), transient ischemic attack (ICD-10 code G45.9), amaurosis fugax (ICD-10 code G45.3), acute myocardial infarction (ICD-10 codes I21.0–4, I21.9, I22.0–1, I22.8–9, I24.8–9), unstable angina (ICD-10 code I20.0) or angina with presence of intervention (ICD-10 codes I20.1 or I20.8-9 together with Z95.1 or Z95.5 or FNG02 or FNG05 or FNC-10, -20, -30, -40, -50, -60 or FNC96) (Appendix A). Complementary data regarding diabetes mellitus and hypertension (defined as systolic blood pressure ≥140 mmHg and/or diastolic blood pressure ≥90 mmHg) were collected from medical records besides diagnoses from the Swedish National Health Register (ICD-10 I10.9, I15.0-2, and I15.8-9).

The Regional Ethics Committee at the University Hospital in Umeå (Dnr 2019-03414, date 21 June 2019) approved the study. The participants gave their written informed consent and it was performed according to the Declaration of Helsinki.

### 2.2. Methods

Enzyme-linked immunosorbent assay (ELISA) was used to detect ACPA (anti-CCP2 antibodies) and performed according to the manufacturer’s instructions (Svar Life Science, Malmö, Sweden). The RF were analyzed using ELISA (Triolab, Mölndal, Sweden) and erythrocyte sedimentation rate (ESR; mm/h) according to Westergren method were analyzed at each hospital. Genotyping of HLA-DRB1 was performed as previously described [33]. HLA shared epitope (SE) was defined as HLA-DRB1*0401/0404/0405/0408/0101.

Radiographs of hands, wrists, and feet were assessed at index date and after 24 months and graded according to the Larsen score method [34] by two experienced readers (E.B. and S.R-D.).

### 2.3. Statistical Analyses

For analyses of categorical data, the chi-square test was used. For comparison of independent continuous variables between two groups, the Student’s *t*-test or the Mann–Whitney U test were used when appropriate. Cox proportional hazard regression analyses were used to take time into account for calculating hazard ratio (HR) and 95% confidence intervals (CI) of hormonal factors for CVE as dependent variable. Area under the curve (AUC) was calculated for cumulated values of DAS28 and ESR over the period of 24 months according to the trapezoid model (DAS28-AUC_24_ and ESR-AUC_24_, respectively) [35]. No significant differences were found among cases who did not respond on the questionnaires and those who did, concerned age at onset, smoking habits, frequency of anti-CCP or RF positivity, DAS28 and ESR at the index date or DAS28-AUC_24_. Since there were missing values in almost all covariates under consideration, we performed a sensitivity analysis using multiple imputation (MI) where missing covariate values were imputed using the other available covariates as well a crude estimate of cumulative baseline hazard as explanatory variables, as suggested by White and Royston [36]. We performed 10 imputations, using the default options in SPSS for determining imputation method for the respective variables as well as for pooling estimates. We did not include the Larsen Score variable in the multiple imputation as the fraction of missing data exceeded 50%. Two-sided *p*-values ≤ 0.05 were considered statistically significant. SPSS software (v. 26.0 IBM Corp, Armonk, NY, USA) was used for the statistical analyses.

## 3. Results

### 3.1. Characteristics of the Patients

Of the 803 postmenopausal female patients with early RA included in the study, a total of 103 patients had one or more CVE. Of them, 39 had their first event before they were diagnosed with or had their first symptoms of RA, mean (SD) 3.5 (5.2) years before RA onset and were thus excluded from further analysis. The other 64 patients had their first CVE after RA diagnosis. In the subsequent analyses, we have focused on comparing patients without CVE with those who had their first CVE after RA onset, since we wanted to focus on events in relation to RA. The total time at risk for first CVE after menopause was 7433 person-years and the incidence rate of first CVE was 7.76/1000 person-years.

Age at RA onset was significantly higher in patients who suffered from CVE compared to those without CVE (Table 1). Mean (SD) age for patients with CVE after RA onset was 59.7 (8.4) years compared with 56.6 (10.6) years in those without CVE (Table 1). DAS28-AUC during the first 24 months was significantly higher in those with CVE after RA onset compared with those without CVE. Traditional risk factors related to CVE such as smoking, hypertension, and diabetes were significantly increased in patients with CVE. Other characteristics including BMI at baseline, HLA-SE, and Larsen score at index date and 24 months, displayed no significant differences between the groups. DAS28-AUC_24_ was significantly higher in the cases affected with a CVE, although there were no differences in treatments with corticosteroids and DMARDs. We did not find significant relationships between age at onset and CVD risk factors, e.g., ever-smoking, or diabetes or DAS-AUC_24_ except for presence of hypertension that was related to age at onset. Regarding treatment, there were no differences between the groups (Table 1).

### 3.2. Hormonal Factors and Association to CVE

In the patients who had their first CVE after RA onset, the mean (SD) time from onset of RA to CVE was 7.4 (4.8) years and the postmenopausal time period was long until CVE was observed, mean time 17.4 years (10.1) (Table 2).

In patients who had a CVE, the time in years from menopause until RA onset was significantly longer (Table 3) and the proportion of postmenopausal patients at RA onset was significantly higher in comparison to those without CVEs. Patients with CVE after RA onset had a higher number of childbirths and were younger at first childbirth. In the group without CVE, the proportion of patients taking oral contraceptives was significantly higher compared with those with CVE. No significant differences were found for age at menarche, menopause or reproductive lifespan between the groups with similar results after multiple imputation (Table 3).

### 3.3. Cox Proportional Hazard Regression Analyses

In simple Cox proportional hazard regression models focusing on CVE after RA onset, higher risk of CVE was associated with age at debut, Larsen score at 24 months, number of children, number of children stratified into 3 groups, menopause (ref. before or after RA onset), years from menopause until RA onset, years from last childbirth to RA onset, and lower age at first childbirth besides having diabetes and hypertension (Table 4). In the multiple variable analyses, age at onset was not included as it correlated with Larsen score 24 months (r_s_ = 0.27, *p* < 0.001), years from menopause to RA onset (r_s_ = 0.904, *p* < 0.001), years from last childbirth (r_s_ = 0.914, *p* < 0.001), and DAS-AUC_24_ (r_s_ = 0.07, *p* = 0.049). In multiple variable analyses, the covariates being significant in simple variable analysis except for age at onset were included; the number of children calculated as continuous variable or stratified into 3 groups, and years from last childbirth remained significantly associated with having a CVE (Table 4a). Significance for diabetes and hypertension was lost. In another multiple variable analysis, the number of children calculated as continuous variable or stratified into 3 groups in separate analyses remained significant (HR = 1.842, 95%CI 1.345, 2.522, *p* < 0.001 and HR = 12.478, *p* = 0.002, 2df, respectively). Smoking, BMI, and use of oral contraceptives did not have a significant association with CVE. Furthermore, the significances for hypertension and diabetes were lost in multiple variable analysis. Analyses after multiple imputations presented in Table 4b yielded similar results except for significant effects of hypertension in the multiple variable analysis including number of children. The factors “Menopause before RA onset” and “Years between menopause and RA onset” were significantly related, whilst calculations performed including only one of the factors yielded similar non-significant results (HR = 1.028, 95%CI 0.414, 2.55, *p* = 0.053 and HR = 1.033, 95%CI 0.987, 1.08, *p* = 0.161, respectively). Further, no significant results were observed in the simple variable analyses of age at menopause, early menopause (≤45 years), reproductive life span, breastfeeding, HRT, progestagen, miscarriage or abortion, ESR-AUC_24_, HLA-SE, anti-CCP antibodies or RF.

## 4. Discussion

In this inception cohort of women with recent onset RA, we have evaluated hormonal and reproductive factors, and the reproductive lifespan in relation to development of CVE. Both RA and CVE are related to hormonal factors, and we have, in addition to Pfeifer et al., as one of the first studies analyzed the impact of these factors for CVE in RA [37].

According to our findings, the convincing majority of cases with later CVE had their RA disease onset after menopause. In nine cases with CVE, RA onset predated menopause, although the mean (SD) age at onset for the nine cases was 54 (4.3) years, suggesting a very late menopause. Patients with CVE after RA onset were significantly older compared with those without a CVE (*p* < 0.05) who had reached menopause. These findings suggest that it is not a long duration of disease burden in RA that account for the increased risk for CVE. Instead, the timing of disease development in relation to hormonal status appears to contribute most to the increased risk. This might be due to a certain vulnerability including aging blood vessels when getting RA after menopause when the hormonal protection of endogenous estrogen exposure is diminished. As reported elsewhere, the incidence of CVD is much lower in premenopausal than postmenopausal women in the general population [38], which we were able to confirm among the cases of this study. It has been suggested that estrogen has both long- and short-term effects on the blood vessel wall mediated by several mechanisms, which can partly explain our findings [39]. Estrogen receptors have been detected in smooth-muscle cells in coronary arteries [40] and data suggest that estrogen affects the bioavailability of endothelium-derived nitric oxide that can cause relaxation of vascular smooth-muscle cells. Thus, the vaso-protective effects of estrogen mediated by a large number of mechanisms could be reduced after menopause when estrogen levels are lower [39].

We were unable to confirm a relationship between reproductive lifespan, early menarche or early menopause and CVE in the present study. Others have reported that early menarche may increase the risk for both ischemic heart disease and stroke in the general population [20]. Early menopause has been associated both with development of RA [11], CVD in RA patients [37], and CVD in the general population [26,41]. A longer reproductive lifespan has been associated with a lower estimated risk of CVD in the general population [25,26,28], which we were unable to confirm. In this study, patients with CVE after RA onset had a slightly higher (18.8%) frequency of early menopause and a slightly shorter reproductive lifespan (mean 36.1 years), although not significantly different compared with RA patients without CVE (15.3% and mean 36.4 years, respectively).

Furthermore, in women of the general population, multiparity and childbirth at an early age increased the risk of CVE, as previously reported by others [20,42,43]. This was also the finding in our study. However, some studies have found a similar risk in men, suggesting that other factors than hormones play a role [21]. According to our findings, adjustments for the traditional risk factors such as smoking, hypertension, diabetes, BMI, and also age, did not reduce the significant results for multiparity and years from last childbirth to RA onset. However, due to lack of data, we have not performed any adjustments for lipid status, social, cultural, psychological, and behavioral factors associated with parenthood that could affect the development of CVEs.

Duration of breastfeeding and frequency of women breastfeeding did not differ between the groups, although we found a tendency for more frequent breastfeeding and longer duration in women with CVE after RA onset. Several others have reported that breastfeeding may decrease the risk for CVD in hospitalization cases or CVD mortality, but our findings did not confirm this [24,43].

In contrary to other studies on the general population [23], we were unable to conclude that oral contraceptives increase the risk of CVE. Controversially, our results showed a tendency towards protective effects of oral contraceptives irrespective of treatment duration. In previous studies on oral contraceptives in RA, a protective effect was found for disease development, particularly in ACPA-positive RA [12]. Furthermore, we have reported in a previous study that disease development of RA was postponed with oral contraceptives [44].

Hormone replacement therapy has been suggested to increase the risk for RA [45], whilst another study reported a decreased risk in ACPA-positive RA [13]. Furthermore, findings in the general population suggest that HRT has negative effects on CVD and CVE, e.g., increased risk for stroke, venous and pulmonary thromboembolism, probably by affecting lipids, lipoproteins, inflammation, and hemostasis [46]. We found no effect of HRT for the risk of CVE, although the levels of these mentioned factors were not analyzed in this study.

We cannot explain the presence of CVE by higher age at disease onset or longer time from menopause to RA onset by the traditional risk factors. Presence of ever-smoking, diabetes or higher disease activity was not related to age, except for hypertension. The RA patients with CVE had significantly higher DAS28-AUC_24_ compared with those without a CVE, although the treatment with corticosteroids and DMARDs were similar in both groups. The significance of DAS28-AUC_24_ was lost when time was taken into account in the Cox proportional hazard regression analyses. A relationship between CVD and inflammation has been demonstrated in other studies, which we were unable to confirm in this study [47,48]. However, this was not our main focus in our study, and the only measurements of inflammation were DAS28-AUC_24_ and ESR-AUC_24._

A strength of the current study was that the cohort of early RA patients was unselected and subsequently included and originated from a homogenous population of northern Sweden. Almost all individuals were willing to participate and to be included in this study (80%), of those diagnosed with early RA and younger than 80 years of age, within the catchment area of northern Sweden. Another strength is that we have chosen a rather strict definition for CVEs and have not included all types of cardio- and cerebrovascular diseases. This strict definition decreases the risk of including patients with other causes for cardio- and cerebrovascular diseases. In contrary to some other studies, patients with other cardio-and cerebrovascular diseases such as congestive heart failure and cerebral hemorrhage were not included, although these conditions might also have been affected by hormonal and reproductive factors and the RA disease. Moreover, in our analyses, we have performed adjustments for traditional risk factors. We could confirm that the RA patients with CVE had higher frequencies of being a smoker, hypertension and/or diabetes mellitus compared with those without an event, although these factors were not significant in multiple variable analyses.

Although this study included a fairly large number of patients, stratification into subgroups resulted in a limited number of cases. Another limitation could be the difficulties in accurately recalling past hormonal events. Even though the agreement has been shown to be high between medical records and interview data regarding the duration and use of oral contraceptives, women might have difficulties in accurately recalling what specific types of preparations they have used [49,50]. Furthermore, age at menopause and menarche can be hard to remember exactly, especially since menopause does not occur at one specific time point. A larger patient material could help to overcome some of the limitations. Another limitation is that we did not have a control group from the general population to compare with. Additionally, more information about CVD risk factors and CVD medications would be informative in the RA cases and in the controls. There may, of course, be residual confounding factors we are unaware of or a lack of data for analysis.

The aim of this study was to evaluate the contribution of hormonal and reproductive factors for the development of CVE in patients with early RA. Some of the factors seem to act similarly for the two diseases with increasing or decreasing the risk, where previous studies have described contradictory effects [11,20]. Number of pregnancies increased the risk for CVE in RA as previously reported for CVD in the population [20,42]. The reproductive lifespan tended to be shorter in our cases with CVE, although not significantly, in line with previous findings on the general population [25,26,27,51]. Early menopause also tended to be of higher frequency, 18.8% vs. 15.3% in RA cases with CVE, but not significantly, whilst pregnancy at an early age was significantly associated with CVE in RA as has been reported for both CVD and RA per se [11,20,22,52]. The risk contribution of oral contraceptives on RA disease onset have been presented with contradictory results previously [11,53,54,55,56], whilst most studies have reported an increased risk for CVD in the general population [23]. It is evident that the hormonal and reproductive factors act both similarly and differently in these two diseases, which makes it difficult to conclude about interactive effects. However, we found that the significance of hypertension and diabetes was lost in multiple variable analyses, whilst the number of childbirths remained.

## 5. Conclusions

In summary, this is one of the first studies exploring the impact of hormonal and reproductive factors for CVE in women affected with RA. Female RA patients with CVE had a late RA disease onset, in most cases, after menopause. A higher number of childbirths increased the risk for CVE, as well as years from last childbirth, irrespective of traditional risk factors. Several other hormonal and reproductive factors showed a trend for increased risk of CVE, such as early menopause, pregnancy at an early age, and reproductive life span, although not significantly. Larger studies are needed to evaluate this further. The DAS28-AUC_24_ was higher in cases with a CVE, although that was lost when time was taken into account. It is evident that the hormonal and reproductive factors act both similarly and differently in RA versus the general population, which makes it difficult to conclude about the interactive effects of these factors in RA and CVE per se. Understanding the role of female hormonal factors is of key importance for the identification of women at high risk for CVE so that preventive strategies can be initiated. These findings help to identify factors that can cause increased mortality and morbidity in women with RA besides disease-related factors and traditional risk factors for CVD.

## Figures and Tables

**Figure 1 jcm-12-00208-f001:**
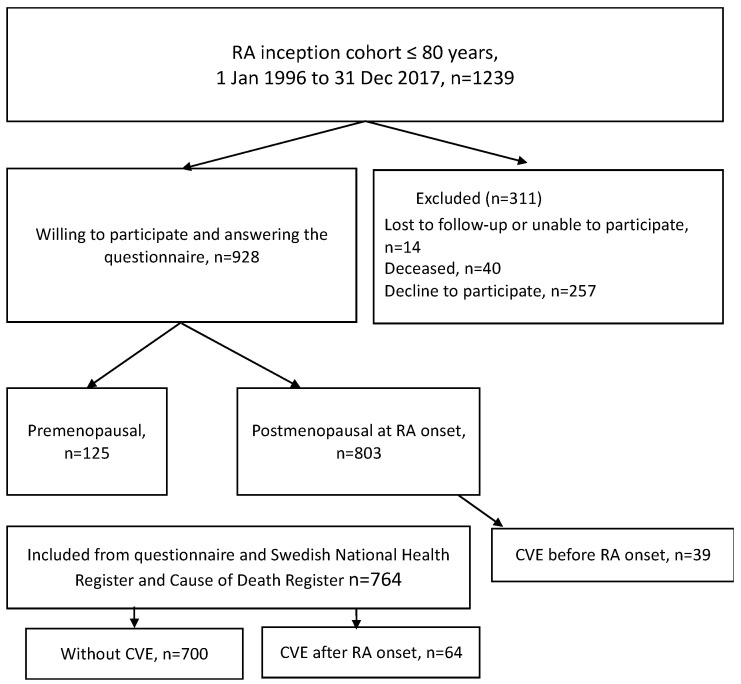
Flowchart illustrating the inception cohort of RA patients.

**Table 1 jcm-12-00208-t001:** Baseline characteristics of RA patients without CVE and CVE after RA onset.

Baseline Characteristics	No CVE	CVE after RA Onset
	(*n* = 700)	(*n* = 64)
Age at RA onset, years, mean (SD)	56.6 (10.6)	59.7 (8.4) *
Follow-up time, years, mean (SD)	10.0 (5.5)	12.9 (4.9) ***
DAS28-AUC_24_, mean (SEM) ^1^	86.0 (0.9)	92.3 (3.0) *
ESR-AUC_24_, mean (SEM) ^2^	510.6 (11.2)	573.9 (40.6)
Ever-smoker, *n* (%)	360/603 (59.7)	45/61 (73.8) *
Hypertension, *n* (%)	114/356 (32.0)	39/64 (60.9) ***
Diabetes mellitus, *n* (%)	24/356 (6.7)	11/64 (17.2) **
BMI, kg/m^2^, mean (SEM) ^3^	26.3 (0.2)	26.6 (0.5)
HLA-SE, *n* (%)	394/550 (71)	45/57 (78.9)
Larsen score baseline, median (IQR) ^4^	5.0 (8.0)	8.0 (11.0)
Larsen score at 24 months, median (IQR) ^5^	9.0 (8.0)	12.0 (12.5)
Radiological progression, yes/no, *n* (%)	70/180 (38.9)	10/29 (34.5)
Anti-CCP2 positivity, *n* (%)	422/610 (69.2)	39/60 (65.0)
RF positivity, *n* (%)	455/598 (76)	43/58 (74.1)
Corticosteroids at baseline, *n* (%)	353/572 (61)	37/61 (60.7)
Corticosteroids still at 24 months, *n* (%)	151/452 (33.4)	17/57 (29.8)
csDMARDs at baseline, *n* (%)	609/675 (90)	58/64 (90.6)
Methotrexate at baseline, *n* (%)	550/669 (82.2)	55/63 (87.3)
csDMARDs ever, *n* (%)	677/680 (99)	63/64 (98.4)
bDMARDs ever, *n* (%)	112/689 (16.3)	6/64 (9.4)
bDMARD within 24 months, *n* (%)	100/689 (14.5)	5/64 (7.8)

* Significance to reference (no CVE) * *p* < 0.05, ** *p* < 0.01, *** *p* < 0.001. Anti-CCP2, anticyclic citrullinated; peptide-2; AUC, area under the curve; bDMARDs, biological disease-modifying antirheumatic drugs; BMI, body mass index; csDMARDs, conventional disease-modifying antirheumatic drugs; DAS, disease activity score; ESR, erythrocyte sedimentation rate; RF, rheumatoid factor. Number of cases with data before imputation ^1^
*n* = 691 and 64, ^2^
*n* = 700 and 64, ^3^
*n* = 608 and 58, ^4^
*n* = 275 and 33, ^5^
*n* = 256 and 31, respectively.

**Table 2 jcm-12-00208-t002:** Characteristics of patients with their first CVE after (*n* = 64) RA onset.

Characteristics of Patients	CVE after RA Onset (*n* = 64)
Age at event, years, mean (SD)	67.1 (8.6)
Duration of RA until CVE, years, mean (SD)	7.4 (4.8)
Number of CVE per individual, mean (SD)	1.2 (0.4)
Individuals at menopause at CVE, *n* (%)	60 (93.8)
Years from menopause until CVE, mean (SD)	17.4 (10.1)
Years from last childbirth until CVE, years, mean (SD)	39.4 (12.0)
TIA, AF or ischemic stroke, *n* (%)	37 (58.7)
MI or AP with intervention, *n* (%)	31 (48.4)
CVE as cause of death after RA onset, *n* (%)	4 (6.3)

AF, amaurosis fugax; AP, angina pectoris; CVE, cardiovascular event; MI, myocardial infarction; TIA, transient ischemic attack.

**Table 3 jcm-12-00208-t003:** Characteristics of hormonal and reproductive factors in patients with no CVE (*n* = 700), and CVE after RA onset (*n* = 64).

Hormonal and Reproductive Factors	No CVE	CVE after RA
	(*n* = 700)	(*n* = 64)
Age at menarche, mean (SD), years	13.2 (1.4)	13.4 (1.4)
Age at menopause, mean (SD), years	49.8 (4.4)	49.5 (5.3)
Early menopause (≤45 years), *n* (%)	107 (15.3)	12 (18.8)
Years from menopause until RA onset, mean (SD)	6.7 (11.2)	10.2 (9.8) *
Individuals at menopause at RA onset, *n* (%)	517 (73.9)	55 (85.9) *
Childbirth before RA onset, *n* (%)	622 (90.3)	61 (95.3)
Number of childbirths, median (IQR)	2.0 (2.0)	2.0 (1.0) *
Age at first childbirth before RA onset, mean (SD)	24.2 (4.7)	22.9 (4.2) *
Age at last childbirth before RA onset, mean, (SD)	29.3 (5.3)	29.1 (5.5)
Years from last childbirth until RA onset, mean (SD), years	27.8 (11.8)	30.8 (10.4)
Miscarriage or abortion, *n* (%)	247 (36.8)	19 (30.2)
Breastfeeding, *n* (%)	572 (92.0)	57 (93.4)
Total duration of breastfeeding months, mean (SD)	12.7 (12.4)	13.4 (12.1)
Oral contraceptives, *n* (%)	438 (63.3)	32 (50.0) *
Oral contraceptives, mean (SD), years	3.7 (5.5)	2.5 (4.2)
Progestagen, *n* (%)	182 (26.7)	14 (21.9)
Progestagen, mean (SD), years	2.0 (5.0)	2.1 (6.4)
HRT before RA onset, *n* (%)	184 (27.6)	16 (25.4)
HRT, mean (SD), years	6.4 (6.3)	6.1 (6.6)
Reproductive lifespan, mean (SD), years	36.4 (5.2) (*n* = 559) ^1^	36.2 (5.5) (*n* = 59) ^1^

* CVE after RA onset vs. reference (no CVD); * *p* < 0.05, CVE, cardiovascular event; HRT, hormone replacement therapy, ^1^ number of included cases.

**Table 4 jcm-12-00208-t004:** (a). Simple and multiple variables in Cox proportional hazard regression analysis of CVE after RA onset as dependent variable (*n* = 64) versus no CVD (*n* = 700). Complete case analyses. (b). Simple and multiple variables in Cox proportional hazard regression analysis of CVE after RA onset as dependent variable (*n* = 64) versus no CVD (*n* = 700). Multiple imputation results.

(a)									
Covariates	Simple Variable Analysis			Multiple Variable Analysis			Multiple Variable Analysis		
	HR	95%CI	*p*-Value	HR	95%CI	*p*-Value	HR	95%CI	*p*-Value
Age at onset, years	1.101	1.067, 1.136	0.000	-			-		
Ever smoker	1.572	0.889, 2.783	0.120						
Larsen score at 24 months	1.035	0.998, 1.074	0.065		.				
Children, *n*	1.366	1.114, 1.675	0.003	1.842	1.345, 2.522	0.000			
Children (3 groups)	8.415		0.015				12.478 ‡		0.002
≤1 child	Ref.								
2 children	1.795	0.890, 3.621	0.102				3.195	1.309, 7.801	0.011
≥3 children	2.816	1.382, 5.735	0.004				5.673	2.147, 14.990	0.000
Menopause before RA onset, yes	1.982	1.024, 3.837	0.042	1.228	0.479, 3.148	0.669	1.336	0.513, 3.480	0.553
Years between menopause and RA onset	1.065	1.036, 1.094	0.000	1.026	0.977, 1.078	0.304	1.034	0.985, 1.084	0.177
Oral contraceptives, yes	0.715	0.438, 1.169	0.181						
Age at first childbirth, years	0.941	0.882, 1.004	0.067	1.013 ‡	0.933, 1.099	0.761	1.010 ‡	0.931, 1.096	0.813
Years from last childbirth to RA onset, years	1.070	1.042, 1.099	0.000	1.067	1.010, 1.126	0.020	1.060	1.004, 1.120	0.034
DAS28-AUC_24m_	1.004	0.994, 1.015	0.431						
Diabetes	2.242	1.166, 4.309	0.015	1.088	0.499, 2.368	0.833	1.210	0.554, 2.642	0.633
Hypertension	2.780	1.659, 4.659	0.000	1.676	0.971, 2.892	0.064	1.635	0.944, 2.832	0.079
BMI	1.024	0.966, 1.085	0.426						
Wald statistic for overall significance test. ‡ Included since it was significant in the multiple imputation analysis.									
**(b)**									
**Covariates**	**Simple Variable Analysis**			**Multiple Variable Analysis**			**Multiple Variable Analysis**		
	**HR**	**95%CI**	***p*-Value**	**HR**	**95%CI**		**HR**		
Age at onset, years	1.101	1.067, 1.136	0.000	-			-		
Ever smoker	1.544	0.869, 2.746	0.139						
Larsen score at 24 months †	1.035	0.998, 1.074	0.065						
Children, n	1.366	1.114, 1.675	0.003	1.670	1.268, 2.200	0.000			
Children (3 groups)	8.415		0.015				12.478		0.002
≤1 child	Ref.								
2 children	1.795	0.890, 3.621	0.102				2.753	1.272, 5.958	0.010
≥3 children	2.816	1.382, 5.735	0.004				4.777	2.003, 11.393	0.000
Menopause before RA onset, yes	3.668	1.901, 7.0771	0.000	1.198	0.473, 3.036	0.473	1.256	0.105, 2.459	0.401
Years between menopause and RA onset	1.077	1.051, 1.105	0.000	1.036	0.988, 1.086	0.146	1.041	0.994, 1.091	0.089
Oral contraceptives, yes	0.702	0.429, 1.148	0.159						
Age at first childbirth, years	0.936	0.878, 0.999	0.047	1.018	0.941, 1.101	0.655	1.019	0.942, 1.101	0.645
Years from last childbirth to RA onset, years	1.066	1.039, 1.094	0.000	1.060	1.006, 1.116	0.028	1.057	1.004, 1.112	0.033
DAS28-AUC_24m_	1.004	0.994, 1.015	0.429						
Diabetes	2.265	1.173, 4.371	0.015	1.427	0.715, 2.849	0.314	1.527	0.764, 3.053	0.231
Hypertension	2.802	1.671, 4.700	0.000	1.732	1.008, 2.976	0.047	1.686	0.979, 2.904	0.060
BMI	1.025	0.967, 1.087	0.404						
Wald statistic for overall significance test. The Wald test is not pooled, i.e., complete case results shown for overall test. † Not imputed as the fraction of missing exceeded 50%.									

## Data Availability

Data will be available after contact with the corresponding author. This cohort of patients are included in a large cohort where statistical work and evaluation is ongoing.

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
