# Peer review of "Hormonal and Reproductive Factors in Relation to Cardiovascular Events in Women with Early Rheumatoid Arthritis"

_jcm, 2022, doi:10.3390/jcm12010208_

Round 1
Reviewer 1 Report
This is an interesting paper. I have some comments:
- the main limitation of the study is the lack of some important cardiovascular risk factors such as dyslipidemia and chronic kidney disease, and of cardio-active medications. These data impair the optimal assessment of cardiovascular risk status
- Many articles have demonstrated an association between acute and chronic inflammation and cardiovascular events (doi: 10.1002/art.39098, doi: 10.1001/jama.2022.11390). The authors should comment a bit more on this in their discussion.
Otherwise, this is a nice and weel-organised paper and English is very good.
Author Response
"Please see the attachment."

Reviewer 2 Report
In this original research article, the authors analysed the impact of interactions between hormonal and reproductive factors correlated with cardiovascular events in early RA women. The topic is relevant and interesting. However, I recommend analysing available scientific literature and modify some aspects in order to improve your work:
In line 28 it is stated that ‘patients with rheumatoid arthritis (RA) have an increased risk of cardiovascular disease (CVD)’. Which are more prevalent? A brief comparative overview of the most impacting CVDs would be relevant.
It is necessary to highlight the pathophysiological mechanisms of RA that are correlated with the development of cardiovascular disease, as well as how hormonal factors are implicated in the development of RA. The following topical work may help to improve the information provided: PMID: 34831081.
It is said that conflicting findings had been found in clinical and experimental investigations addressing the pathogenic and etiological significance of female hormones in RA. Which direction do the authors opt for more? It is worth mentioning.
In line 46 it is stated that ‘Several hormonal and reproductive factors in women…’. It is important to include these factors for better clarity. I suggest checking and referring to PMID: 29167771
Cardiovascular events in early RA are a result of inflammatory load. In addition, DMARDs have been shown to reduce cardiovascular risk, so it is worth mentioning several issues related to treatment in RA and factors that may influence cardiovascular risk (drug interactions in RA therapeutic management, CVDs, hormonal issues, etc.). The following topical work may help to improve the information regarding the therapeutic management of RA: PMID: 36058148.
A clearer and more detailed aim needs to be stated in the introduction, the information is too briefly presented.What is the novelty/special aspects that your study brings to the field?
Subsections in the section 2 must be numbered.
Graphical part is missing in the manuscript.
Section 2, for 2.1. Subjects' selection: I suggest a flow chart (CONSORT) describing the criteria for patient's selection.
In line 81 certolizumab pegol should be written in two words. In line 119 it is necessary to briefly describe those ‘routine methods’ for better clarity.
Tables 1 to 3. Head of the table must be completed for the 1st column as well.
How could the limitations of the study be addressed in a future research direction?
The Conclusions section needs to be improved because it concludes far fewer issues than were assessed in the study. Highlight the importance/novelty of your data.
Author Response
In this original research article, the authors analysed the impact of interactions between hormonal and reproductive factors correlated with cardiovascular events in early RA women. The topic is relevant and interesting. However, I recommend analysing available scientific literature and modify some aspects in order to improve your work:
- In line 28 it is stated that ‘patients with rheumatoid arthritis (RA) have an increased risk of cardiovascular disease (CVD)’. Which are more prevalent? A brief comparative overview of the most impacting CVDs would be relevant.
Reply: RA patients are associated with an increased risk of cardiovascular morbidity and mortality because numerous cardiac structures are involved in the pathological processes in particular ischemic heart disease. Other cardiovascular manifestations are
atherosclerosis, arterial stiffness, coronary arteritis, congestive heart failure,
and valvular disease. We have added information to the text, line 30 and also added two references (Roman, M.J.; Salmon, J.E. Circulation 2007, 116, 2346–2355 and Nicola PJ, et al., Arthritis Rheum. 2005;52:412– 420)
- It is necessary to highlight the pathophysiological mechanisms of RA that are correlated with the development of cardiovascular disease, as well as how hormonal factors are implicated in the development of RA. The following topical work may help to improve the information provided: PMID: 34831081
Reply: A number of risk factors are in common between development of RA and of CVD per se such as smoking, air pollution, over-weight, increased BMI, increased CRP, ACPA as summarized in a review (Radu AF, Bungau SG. Management of Rheumatoid Arthritis: An Overview. Cells. 2021 Oct 23;10(11):2857. doi: 10.3390/cells10112857. PMID: 34831081; PMCID: PMC8616326.).
This information has been added to the “Introduction” section.
The increased prevalence of RA in women suggests that female hormonal factors play a role in the development of the RA disease.
Interestingly, estrogens have both pro- and anti-inflammatory effects that may explain their paradox in RA pathogenesis. While estrogens are potent inducers of B cell survival and antibody production, they also decrease the production of key cytokines for RA pathogenesis, such as IL-6, TNFα, and IL-1β (Wong LE, Huang WT, Pope JE et al. Effect of age atmenopause on disease presentation in early rheumatoid arthritis: results from the Canadian Early Arthritis Cohort. Arthritis Care Res 2015;67:61623])
The peak incidence of onset in females coincides with menopause, when the ovarian production of sex hormones drops markedly (Kvien, T.K., Uhlig, T., Odegard, S., Heiberg, M.S., 2006. Epidemiological aspects of rheumatoid arthritis: the sex ratio. Ann. N. Y. Acad. Sci. 1069, 212e22)
Other hormones, such as androgens, progesterone or prolactin, exert also effects on the immune system. The post-menopause stage, an early age at menopause, the post-partum period and the use of anti-oestrogen agents are associated with RA onset. All these phenomena have in common an acute decline in ovarian function and/or in oestrogen bioavailability. (Alpizar-Rodriguez, D., Pluchino, N., Canny, G., Gabay, C., and Finckh, A. The role of female hormonal factors in the development of rheumatoid arthritis. Rheumatology (Oxford) 2017; 56, 1254-1263).
We have added information to the text, line 50.
3.It is said that conflicting findings had been found in clinical and experimental investigations addressing the pathogenic and etiological significance of female hormones in RA. Which direction do the authors opt for more? It is worth mentioning.
Reply: We have added information about aestrogens as well as other hormones to the text in the “Introduction” section. We are not quite sure what the reviewer exactly mean by the comment.
- In line 46 it is stated that ‘Several hormonal and reproductive factors in women…’. It is important to include these factors for better clarity. I suggest checking and referring to PMID: 29167771
Reply: We have tried to reorganize the presentation in the text (line 67 to 71) to make the effects of hormonal and reproductive factors more evident in the general population.
- Cardiovascular events in early RA are a result of inflammatory load. In addition, DMARDs have been shown to reduce cardiovascular risk, so it is worth mentioning several issues related to treatment in RA and factors that may influence cardiovascular risk (drug interactions in RA therapeutic management, CVDs, hormonal issues, etc.). The following topical work may help to improve the information regarding the therapeutic management of RA: PMID: 36058148.
Reply: The reviewer points that cardiovascular events in early RA are a result of inflammatory load. This is not completely settled as there are other risk factors in common for RA and CVD, as well (smoking, air pollution, over-weight, increased BMI, increased CRP etc). We have recently published a manuscript where we found that individuals without measurable inflammation but who subsequently developed RA had already years before symptom onset increased frequencies of risk factors for CVD. After disease onset the RA cases had significantly increased frequency of CVE in early RA compared with matched controls (Kokkonen et al . Cardiovascular risk factors predate the onset of symptoms of rheumatoid arthritis: a nested case-control study. Arthritis Res Ther. 2017 Jun 30;19(1):148. doi: 10.1186/s13075-017-1351-8. And Kokkonen et al Hormonal and reproductive factors in relation to cardiovascular events. Accepted JCM Dec 2022)).
Atherosclerosis is a chronic inflammatory disease with a probable genetic component. If inflammation is an important factor in the development of cardiovascular disease, then effective control of inflammation by pharmacological interventions would be expected to diminish the risk of cardiovascular disease development. Unfortunately, despite the considerable improvements in disease control in the past couple of decades, including during the early phases of disease, the incidence of cardiovascular disease still remains elevated in RA.
Different modifications of drugs such as NSAIDs, will increase the risk for CVD and corticosteroids, have also pro-atherogenic effects. Conventional synthetic DMARDs, biological DMARDs and target synthetic DMARDs have all effects on inflammation but they have different effects on glucose metabolism, blood pressure, cholesterol metabolism and on thrombosis and coagulation. This has been extensively analysed in a recent publication
( ref., Atzeni F, et al., Nat Rev Rheumatol. 2021 May;17(5):270-290. doi: 10.1038/s41584-021-00593-3. Epub 2021 Apr 8. Erratum in: Nat Rev Rheumatol. 2021 Apr 16;: PMID: 33833437).
In our study we found that DAS28-AUC24 was significantly higher in the cases affected with a CVE, although there were no differences in treatments with corticosteroids and DMARDs. In Cox proportional hazard regression analysis when time were taken into account the significance of DAS28-AUC24 as covariate was lost.
We have added a sentence to the “Results” section, 186-188 and to the “Discussion” section, line 374-381.
6.A clearer and more detailed aim needs to be stated in the introduction, the information is too briefly presented. What is the novelty/special aspects that your study brings to the field?
Reply: We have as one of the first studies focused on factors previously analysed for women of the general population to explore the impact of these factors in women with RA for affection with CVE. We have tried to improve our aim of the study by expressing more clearly the aim.
7.Subsections in the section 2 must be numbered.
Reply: The subsections have been numbered in the section 2.
- 8. Graphical part is missing in the manuscript.
Reply: Originally, we have not included any graphical part in the manuscript, although we have now added a flow chart of the patient material.
- Section 2, for 2.1. Subjects' selection: I suggest a flow chart (CONSORT) describing the criteria for patient's selection.
Reply: We have now added numbers for section 2 as the reviewer points out.
10.In line 81 certolizumab pegol should be written in two words.
Reply: Corrected in agreement with the comment
11.In line 119 it is necessary to briefly describe those ‘routine methods’ for better clarity.
Reply: The RF were analysed using ELISA (Triolab, Mölndal, Sweden) and erythrocyte sedimentation rate (ESR; mm/h) according to Westergren method were analysed at each hospital. The sentence has been corrected.
12.Tables 1 to 3. Head of the table must be completed for the 1st column as well.
Reply: Corrected and information added to the first column
Table 1: Baseline characteristics
Table 2: Characteristics of patients
Table 3. Hormonal and reproductive factors
- How could the limitations of the study be addressed in a future research direction?
Reply: Larger patient material would increase the possibility to stratify the cases into subgroups. Further a control group from the general population to compare with would be wished for. We also would like to have more information about CVD factors and CVD medications in the RA cases and controls. Sentences been added to the “Discussion” section.
- The Conclusions section needs to be improved because it concludes far fewer issues than were assessed in the study. Highlight the importance/novelty of your data.
We have tried to rewrite the “Conclusion” section as the reviewer suggests.

Round 2
Reviewer 2 Report
Figure 1 must be revised. Only text with no frame? Please check a CONSORT type flow chart. The title of the figure must be under the figure, not above.
L134. Please provide the no/date of the ethical approval.
Author Response
Please see the attachment."
